# Association of Urinary Lead and Cadmium Levels, and Serum Lipids with Subclinical Arteriosclerosis: Evidence from Taiwan

**DOI:** 10.3390/nu15030571

**Published:** 2023-01-21

**Authors:** Chien-Yu Lin, Sandy Huey-Jen Hsu, Ching-Way Chen, Chikang Wang, Fung-Chang Sung, Ta-Chen Su

**Affiliations:** 1Department of Internal Medicine, En Chu Kong Hospital, New Taipei City 237, Taiwan; 2School of Medicine, Fu Jen Catholic University, New Taipei City 242, Taiwan; 3Department of Environmental Engineering and Health, Yuanpei University of Medical Technology, Hsinchu 300, Taiwan; 4Department of Laboratory Medicine, College of Medicine, National Taiwan University Hospital, National Taiwan University, Taipei 100, Taiwan; 5Department of Cardiology, National Taiwan University Hospital, Yunlin Branch, Yunlin 640, Taiwan; 6Department of Health Services Administration, College of Public Health, China Medical University, Taichung 404, Taiwan; 7Department of Food Nutrition and Health Biotechnology, Asia University, Taichung 413, Taiwan; 8Department of Environmental and Occupational Medicine, National Taiwan University Hospital, Taipei 100, Taiwan; 9Department of Internal Medicine and Cardiovascular Center, National Taiwan University Hospital, Taipei 100, Taiwan; 10Institute of Environmental and Occupational Health Sciences, College of Public Health, National Taiwan University, Taipei 100, Taiwan; 11The Experimental Forest, National Taiwan University, Nantou 558, Taiwan

**Keywords:** lead, cadmium, cardiovascular diseases, carotid intima-media thickness, low-density lipoprotein cholesterol, small dense low-density lipoprotein cholesterol

## Abstract

Background: Exposure to lead and cadmium has been linked to changes in lipid metabolism and the development of arteriosclerosis, but the role of lipoprotein profiles in this relationship is not well understood, including the potential role of novel lipid biomarkers. Methods: In this study, we enrolled 736 young Taiwanese subjects aged 12 to 30 years to assess the correlation between urine levels of lead and cadmium, lipoprotein profiles, and carotid intima-media thickness (CIMT). Results: Higher levels of lead and cadmium were significantly associated with higher levels of low-density lipoprotein cholesterol (LDL-C), small dense LDL-C (sdLDL-C), LDL-triglyceride (LDL-TG), and CIMT. Participants with higher levels of lead and cadmium had the highest mean values of CIMT, LDL-C, sdLDL-C, and LDL-TG. In a structural equation model, lead had a direct and indirect association with CIMT through LDL-C and sdLDL-C, whereas cadmium had a direct association with CIMT and an indirect association through LDL-C. Conclusion: Our results suggest higher levels of lead and cadmium are associated with abnormal lipid profiles and increased CIMT. These heavy metals could have additive effects on lipids and CIMT, and the relationship between them may be mediated by lipoprotein levels. Further research is needed to determine the causal relationship.

## 1. Introduction

Cardiovascular diseases (CVDs) are currently the main cause of mortality in humans, although great progress has been made in their treatment [1]. In addition to traditional risk factors such as diabetes mellitus and smoking, environmental lead and cadmium exposure have recently been identified as crucial CVD risk factors [2,3]. Lead and cadmium have been identified as two of the top ten chemicals of major public health concern by the World Health Organization (WHO) [4]. Lead and cadmium exposure can increase oxidative stress, resulting in toxic effects on lipids, proteins, and DNA molecules. As a result, exposure to these two heavy metals can have adverse effects on various organ systems, including CVDs [5]. In our previous study of a young Taiwanese population, we showed that higher levels of urine lead and cadmium levels were correlated with a greater carotid intima-media thickness (CIMT), a biomarker of subclinical arteriosclerosis [6], and endothelium–platelet microparticles, which are associated with endothelial damage [7]. Being alert to such danger, human exposure to lead and cadmium has been decreasing for decades [8,9]. However, exposure to low doses of lead and cadmium are still associated with adverse health outcomes in recent epidemiological studies [10,11].

The lipids that accumulate beneath vascular endothelial cells eventually cause the formation of atherosclerotic lesions [12]. Many lipoprotein biomarkers have been proven to predict CVD morbidity and mortality [13]. The most frequently used parameters are low-density lipoprotein cholesterol (LDL-C), high-density lipoprotein cholesterol (HDL-C), and triglycerides. Other new biomarkers of lipoprotein, including small dense LDL-C (sdLDL-C), low-density lipoprotein triglyceride (LDL-TG), and lipoprotein (a), were also proven to predict CVD events in recent studies [7,14,15].

Previous experimental studies have shown that exposure to lead or cadmium can affect lipid metabolism [16,17]. However, the results of high-dose exposure in the laboratories cannot be used to infer the results of low-dose exposure in everyday life. For lead, the Centers for Disease Control and Prevention (CDC) considers a blood lead level of 5 micrograms per deciliter (µg/dL) or higher to be elevated [18]. For cadmium, the WHO has established a reference level of 2.5 µg/L in blood as an indicator of long-term exposure to cadmium [19]. In epidemiology, researchers have explored the relationship between lead or cadmium and serum lipid profiles. However, the lipid markers used in these studies are limited to traditional lipid biomarkers, such as total cholesterol, LDL-C, non-HDL-C, and triglyceride. These studies all show a consistent, positive association between lead and lipid profiles, whereas the relationship between cadmium and lipids is inconsistent among studies [20,21,22]. Additionally, only one previous report investigated the synergistic effect of lead and cadmium on lipids and did not find any additive effect [20].

In summary, previous studies have shown that exposure to lead or cadmium is associated with subclinical arteriosclerosis and endothelial damage. Although we have known these two heavy metals have been shown to negatively affect lipid metabolism, and some lipoprotein biomarkers have been linked to an increased risk of CVDs, there has not yet been a study that specifically examines the role of lipid profiles in the relationship between these heavy metals and arteriosclerosis. In addition, the relationship between these heavy metals and novel lipoprotein biomarkers has not yet been explored. Furthermore, as humans are often exposed to both lead and cadmium simultaneously, it would be valuable to investigate the effects of co-exposure on lipoprotein profiles and arteriosclerosis. To answer the above questions, we measured urine levels of lead and cadmium, biomarkers of lipid profiles (LDL-C, sdLDL-C, LDL-TG, HDL-C, lipoprotein (a), apolipoprotein A1, apolipoprotein B, and triglyceride), and carotid intima-media thickness (CIMT) in the Young Taiwanese Cohort (YOTA) study.

## 2. Materials and Methods

### 2.1. Study Population and Data Collection

From 2006 to 2008, the YOTA study followed up with students who had undergone abnormal urine screening years earlier and were divided into two groups based on their blood pressure status: elevated or normal, according to the American Heart Association criteria [23]. The students were invited to participate in health checks, with higher participation among those who had elevated blood pressure in childhood [24]. After informed consent was obtained from each participant, cardiovascular health examinations and structured questionnaires were carried out. This study was approved by the Research Ethics Committee of National Taiwan University Hospital, and all subjects signed the consent form. Of the 886 subjects, 148 subjects were excluded due to a lack of urine samples. Two more subjects were also excluded due to missing information on household income. Finally, this study had a total of 736 participants. Full details are available in the Appendix A.

### 2.2. Measurement of Urinary Lead and Cadmium Levels

The full details of the method used in this study can be found in our previously published articles [25,26]. After thawing the urine samples, 1 mL of the samples were diluted 10-fold with 9 mL of 1% (*v*/*v*) nitric acid (J.T. Baker Chemical Company, Phillipsburg, NJ, USA) in 15 mL polypropylene tubes and analyzed by inductively coupled plasma-mass spectrometry (ICP-MS, 7700 series; Agilent Technologies, Inc., Santa Clara, CA, USA). The lowest detectable levels for lead and cadmium were 0.007 μg/L and 0.006 μg/L, respectively. Additional information can be found in the Appendix A.

### 2.3. Measurement of CIMT

The measurement of CIMT is determined by the distance from the leading edge of the first echogenic line (i.e., the boundary between the lumen and the inner lining of the vessel) to the leading edge of the second echogenic line (i.e., the boundary between the middle layer of the vessel and the outer layer) in the far wall of the vessel. An experienced technician used a high-resolution B-mode ultrasonography (GE Vivid ultrasound system, Horten, Norway) equipped with a 3.5–10 MHz real-time B-mode scanner to examine the CIMT of extracranial carotid arteries; we then applied a software package for vascular ultrasound for off-line automatic calculations. The method used in this study was previously described in our article [27], and the full details can be found in the Appendix A.

### 2.4. Measurement of Lipid Profiles

LDL-C, HDL-C, and triglyceride were measured using enzymatic assays with an autoanalyzer (Toshiba FR-200 automatic chemistry analyzer, Toshiba, Tokyo, Japan). sdLDL-C and LDL-TG were analyzed by automated detergent-based homogeneous methods (Denka Seiken, Tokyo, Japan) on a Toshiba FR-200 automatic chemistry analyzer (Toshiba, Tokyo, Japan). Lipoprotein (a), apolipoprotein A1, and apolipoprotein B1 were estimated by an immunoturbidimetric method used by the COBAS INTEGRA systems (Roche Diagnostics Inc., Indianapolis, IN, USA). All lipoprotein profiles are expressed as mg/dL.

### 2.5. Covariates

Covariates for this study included age (continuous), gender (categorical), household income (categorical), hypertension (categorical), type 2 diabetes mellitus (categorical), smoking status (categorical), alcohol consumption (categorical), exercise (categorical), dietary sweets (categorical), dietary fat (categorical), z score of body mass index (BMI) (continuous), systolic blood pressure (continuous), and homeostasis model assessment of insulin resistance index (HOMA-IR) (continuous). The full details are given in the Appendix A.

### 2.6. Statistical Analysis

Creatinine levels in urine were used to normalize for variations in urine flow in this study, as the concentration of heavy metals in urine can be affected by the rate of urine output [28]. We tested the β coefficients of lipoprotein profiles and CIMT with a one-unit increase in ln-urine heavy metal concentrations in multiple linear regression analysis. We also analyzed the regression coefficients of CIMT when the lipoprotein profiles increased by one unit. We included age, sex, BMI z score, smoking status, alcohol consumption, exercise and household income, dietary sweets, dietary fat, systolic blood pressure, and HOMA-IR as covariates in the analysis. Lead and cadmium were modeled separately, and they were also modeled together in a composite analysis. We also analyzed the β coefficients for CIMT with a one-unit increase in ln-urine heavy metal levels at higher or lower levels of lipoprotein parameters (cut at the 50th percentile) and we calculated the cross-product terms to estimate the interaction in separate and composite analyses.

We applied structural equation modeling (SEM) to explore the relationships between urine heavy metal levels, lipoprotein profiles, and CIMT. We assumed that CIMT was modified by heavy metal exposure directly and indirectly through the lipoprotein profiles. The covariates were the same as in the multiple linear regression analysis. We used the natural log transformation for lead, cadmium, and HOMA-IR due to deviation from the normal distribution. A *p* value < 0.05 was considered significant.

## 3. Results

Among the 736 subjects enrolled in this study, 444 were women and 292 were men, with a mean age of 21.3 years. Descriptive characteristics of the participants are listed in Table 1 (categorical variables) and Appendix A (continuous variables). The geometric means of the lead- and creatinine-adjusted lead levels were 2.20 μg/L and 1.51 μg/g creatinine, respectively. The geometric means of the cadmium and creatinine-adjusted cadmium levels were 0.93 μg/L and 0.63 μg/g creatinine, respectively. None of the people in this study took lipid-lowering medications. The association between urine heavy metal concentrations, lipoprotein profiles, and CIMT is shown in Table 2. A one-unit increase in ln-lead and cadmium concentrations was positively associated with LDL-C, sdLDL-C, LDL-TG, and CIMT. The relationship of CIMT with a unit increase in lipoprotein profiles is shown in Appendix A. One unit increases in LDL-C, sdLDL-C, and apolipoprotein B were positively associated with CIMT. There was no correlation between the other lipoprotein profiles and CIMT.

The interaction between lead and cadmium associated with CIMT, LDL-C, sdLDL-C, and LDL-TG is shown in Table 3. Since the levels of urine lead and cadmium were highly correlated with each other (correlation coefficients 0.816 (*p* < 0.001)), most of the subjects were in the highest and lowest quartiles, with only 55 people in each of the two quartiles in the middle. When using participants with lead ≤ 50th and cadmium ≤ 50th percentiles as a reference, those with lead >50th and cadmium > 50th percentiles had the highest mean values of CIMT, LDL-C, sdLDL-C, and LDL-TG. In Appendix A, we show the linear regression coefficients of CIMT with a unit increase in ln-urine heavy metal concentrations at different concentrations (cut at the 50th percentile) of lipoprotein profiles and interactions between lipoprotein markers and heavy metals. The levels of LDL-C and sdLDL interacted with both heavy metals in the association between heavy metals and CIMT in the separate analyses. In the composite analysis, the levels of LDL-C and sdLDL interacted with the lead levels in the correlation between lead and CIMT, where there was no interaction between the lipoprotein markers and cadmium in the association between cadmium and CIMT.

We show the SEM for heavy metals, lipoprotein profiles, and CIMT in Table 4. In the separate analyses, the two heavy metal levels were positively associated with LDL-C and sdLDL-C, whereas the two lipoprotein profiles were positively associated with CIMT. Higher concentrations of these two heavy metals were also correlated with a greater CIMT. In the composite analysis, urine lead and cadmium were associated with increasing LDL-C. However, only lead, but not cadmium, was positively associated with sdLDL-C. These two lipoprotein profiles were associated with elevated CIMT, and both heavy metals were positively correlated with CIMT. In summary, each heavy metal was modeled separately in the SEM. Lead and cadmium had a direct and indirect association with CIMT through the effect of LDL-C and sdLDL-C. When the two heavy metals were modeled together in the SEM, lead had a direct association with CIMT and an indirect association with CIMT through the effect of LDL-C and sdLDL-C, whereas cadmium had a direct association with CIMT and only an indirect association with CIMT through the effect of LDL-C.

The correlations of these heavy metals with CIMT in the SEM are shown in Table 5. Among the two lipoprotein profiles that have roles as mediators, LDL-C has the strongest indirect effect on the association between the two heavy metals and CIMT in the separate analyses. In the composite analysis, sdLDL-C had the strongest indirect effect on the association between lead and CIMT, whereas LDL-C had the strongest indirect effect on the association between cadmium and CIMT.

## 4. Discussion

Among the lipid profiles we analyzed, we found that higher levels of urine lead and cadmium are correlated with higher levels of LDL-C, sdLDL-C, and LDL-TG. We also found that these two heavy metals have additive effects on LDL-C, sdLDL-C, LDL-TG, and CIMT. Moreover, when these two heavy metals are analyzed separately, they exhibit a direct correlation with CIMT and an indirect association through the effect of LDL-C and sdLDL-C. When we considered these two heavy metals together, the statistical results were similar to those of the separate analysis, except that sdLDL-C had no role in the relationship between cadmium and CIMT. This is the first epidemiological report to explore the relationship of lead or cadmium exposure with many important lipid profiles. Additionally, we provided the first evidence of the complex interactions among two heavy metal co-exposures, lipid profiles, and markers of arteriosclerosis in a young Taiwanese population. These results may be causal because of the long half-life of lead and cadmium and the strong correlation we observed. If such, exposure prevention should begin as early as possible to reduce harm to cardiovascular health.

Recently, the CDC lowered children’s reference blood lead concentrations from 5.0 µg/dL to 3.5 µg/dL [29]. However, there is no safe value of blood lead [18]. In the current cohort composed of a young Taiwanese population, we found higher geometric means of urine lead levels (1.50 µg/g creatinine) than the levels in other countries [30,31]. Additionally, the geometric means of urine cadmium levels (0.63 μg/g creatinine) were also higher than the levels reported in other countries [32,33] and exceeded 0.5 µg/g creatinine, a threshold value thought to start causing kidney damage [19,34]. A previous study reported that Taiwanese people may have higher heavy metal exposure than people from the Western world [35]. Another possible explanation is selection bias. Our study participants were based on those with abnormal urine tests in their childhood. It is possible that this group of people was exposed to higher concentrations of heavy metals, which caused these abnormal health conditions.

Lipid disturbance has been found to play an important role in the pathway of heavy metal-associated CVD [36]. LDL-C is a well-documented CVD risk factor indicator, and LDL-lowering drug therapy significantly reduces the incidence of CVD [37]. LDL particles are a heterogeneous population with different subtypes, and sdLDL-C is considered to be a lipoprotein indicator with greater atherogenic potential than the other subtypes [38]. Hypertriglyceridemia increased the production of LDL-TG, and LDL-TG was also reported to increase CVD events in a prospective study [14]. Lipoprotein (a) is a variant of LDL that contains apolipoprotein B-100 covalently bound to apolipoprotein (a). Recent studies have documented the role of lipoprotein (a) as a risk factor for CVD [15]. Apolipoprotein A1 is the major apolipoprotein content of HDL, whereas apolipoprotein B represents the major component of LDL apolipoprotein. The apolipoprotein B/apolipoprotein A1 ratio has been proven to predict CVD [39]. In our current study, we only observed a positive association between LDL-C, sdLDL-C, and apolipoprotein B with CIMT. It is worth noting that this is a cross-sectional study and the population we studied is relatively young, which could potentially impact these findings. Although CIMT is widely recognized as a strong predictor of future cardiovascular events and a marker of subclinical arteriosclerosis, it is important to note that it is not a direct measure of the disease and may not always accurately reflect its presence or severity [40].

The mechanism of abnormal lipid metabolism caused by lead exposure is mainly related to its toxicity to the liver [36]. In animal studies, lead may alter the metabolism of cytochrome p450, activate cholesterol synthetase, and decrease antioxidant ability [41,42]. In addition, cadmium exposure increases liver fatty acid synthesis and decreases lipoprotein lipase activity [43]. Several epidemiological studies have explored lead and cadmium exposure and dyslipidemia in the general population. One study enrolling 7457 adults from the NHANES 2005–2016 database analyzed the association between blood lead/cadmium concentrations and lipid markers, including total cholesterol, LDL-C, non-HDL-C, apolipoprotein B, and triglycerides. The geometric means (95% CIs) of blood lead and cadmium concentrations were 1.23 (1.21, 1.25) μg/dL and 0.36 (0.35, 0.37) μg/L, respectively. Higher concentrations of blood lead were correlated with higher concentrations of serum lipid profiles (total cholesterol, LDL-C, non-HDL-C, and apolipoprotein B). However, blood cadmium did not show any associations [20]. In another study using data from NHANES 2009 to 2012, a total of 19,591 individuals 0–80 years old were surveyed for a correlation between blood heavy metals and lipid profiles (total cholesterol, LDL-C, serum triglyceride, HDL-C). Increased blood lead and cadmium levels were significantly associated with an increased risk of high total cholesterol, whereas increased blood lead was also associated with LDL-C [21]. Additionally, 2591 Korean adults were included to analyze the associations between heavy metals (blood lead, urinary cadmium) and serum lipid profiles (total cholesterol, triglyceride, LDL-C, non-HDL-C). Higher concentrations of blood lead were found in the dyslipidemia group of total cholesterol, LDL-C, non-HDL-C, and triglycerides, whereas higher urine cadmium levels were found in the dyslipidemia group of LDL-C and non-HDL-C [22]. In the current study, we showed that both urine lead and cadmium concentrations were positively associated with lipid profiles. The inconsistent results among these studies may be due to differences in the underlying basic demographics of the sample subjects, size of the cohorts, specimens, and covariates. Since past reports have never investigated the association between lead/cadmium exposure and new lipoprotein biomarkers, we present the first report that urine levels of these two heavy metals have a positive correlation not only with LDL-C but also with sdLDL-C and LDL-TG.

Arteriosclerosis is a multifactorial disease characterized by lipid accumulation, inflammation, and endothelial dysfunction of the arterial walls [44]. The accumulation of lipids induces endothelial cell apoptosis, which activates leukocytes to produce cytokines and chemokines, resulting in the formation of atheroma [45,46]. In addition to altering lipid metabolism, lead and cadmium exposure can increase oxidative stress, resulting in toxic effects on proteins and DNA molecules. As a result, exposure to these two heavy metals can have adverse effects on various organ systems, including vascular endothelial cells [5]. In addition to increased oxidative stress and inflammation, the expression of matrix metalloproteinases and the initiation of amyloid angiopathy are also possible mechanisms of lead- and cadmium-induced arteriosclerosis [47]. However, no epidemiological study has investigated the role of lipoprotein concentrations in the association between heavy metals and arteriosclerosis. In the current study, in addition to the direct effect, we reported the first evidence that LDL-C and sdLDL-C mediate the relationship between lead or cadmium exposure and arteriosclerosis. However, according to our research results, this indirect effect is not as strong as the direct effect. LDL-C and sdLDL-C might be involved in the process of lead/cadmium-induced arteriosclerosis but do not play a major role. Our results showed that lead/cadmium exposure has a positive association with LDL-TG. However, LDL-TG is not positively correlated with CIMT. It is possible that LDL-TG is not as atherogenic as LDL-C and sdLDL-C and is not involved in the pathophysiological process of lead/cadmium-induced arteriosclerosis.

In the SEM of these two heavy metals, the adjusted β between sdLDL-C and CIMT was higher than the adjusted β between LDL-C and CIMT in both separate and composite analyses. It is possible that sdLDL-C has a more direct effect on arteriosclerosis. However, the associated effect of the two heavy metals on LDL-C is much greater than that on sdLDL-C, so the overall indirect effect of LDL-C in the mediation between the heavy metals and CIMT is higher than that of sdLDL-C. Moreover, the role of sdLDL-C in the association between urine cadmium and CIMT was evident only in the separate analyses, not in the composite analysis. Since the levels of lead and cadmium are highly correlated (correlation coefficient 0.816), the association between cadmium and sdLDL-C may be a secondary correlation instead of a true correlation. It is also possible that the correlation between lead and sdLDL-C is much stronger than the effect of cadmium, making the correlation between cadmium and sdLDL-C insignificant. However, basic research on the effect of lead/cadmium on these novel lipoprotein biomarkers has not been performed, and we do not know the detailed mechanism of the different effects of the two heavy metal exposures on different lipoprotein profiles.

Because the pathway of lead/cadmium-induced cell damage has a similar molecular mechanism, low-dose co-exposure to these two heavy metals may have additive effects. One animal report showed that lead and cadmium have synergistic effects on serum lipid profiles and oxidative stress biomarkers [48]. In epidemiologic studies, co-exposure to lead and cadmium has been reported to aggravate renal tubular dysfunction [10] and blood pressure [49]. However, one epidemiological study did not find an additive effect of lead and cadmium on dyslipidemia [20]. In the present study, we showed the possible synergistic effect of low-dose cadmium and lead on novel lipoprotein biomarkers and subclinical arteriosclerosis.

This study has the following limitations. First, our findings cannot infer a causal relationship because of the cross-sectional study design. Second, our study population was limited to young Taiwanese individuals, so the conclusions cannot be extended to other age groups and races. Third, this study did not analyze other contaminants that were co-exposed along with lead and cadmium and might have been correlated with the lipid profile and CIMT at the same time.

## 5. Conclusions

Our research showed that both lead and cadmium levels in urine were correlated with several lipid profiles and CIMT. Additionally, we found that the effects of these two heavy metals on lipid profiles and CIMT were additive. We also observed complex interactions between co-exposure to these two heavy metals, various lipid profiles, and markers of arteriosclerosis. Further investigation is necessary to determine if these findings are causal and, if so, it would be important to implement exposure prevention as early as possible.

## Figures and Tables

**Table 1 nutrients-15-00571-t001:** Demographic data and metal exposure variables in the studied Taiwanese population (categorical variables) (*n* = 736).

		Numbers (%)
Age (year)	12–19	234 (31.8)
	20–30	502 (68.2)
Gender	Male	292 (39.7)
	Female	444 (60.3)
BMI (kg/m^2^)	<24	571 (77.6)
	≥24	165 (22.4)
Smoking	Active	122 (16.5)
	Not active	614 (83.4)
Drinking	Current	69 (9.4)
	Not-current	667 (90.6)
Household income	<50,000 TWD	285 (38.7)
	≥50,000 TWD	451 (61.3)
Hypertension	Yes	60 (8.2)
	No	676 (91.8)
Diabetes mellitus	Yes	14 (1.9)
	No	722 (98.1)

Abbreviations: BMI: body mass index.

**Table 2 nutrients-15-00571-t002:** Linear regression coefficients (standard error) of lipid profiles and CIMT with a unit increase in ln-urine heavy metal concentration (µg/g creatinine) in the multiple linear regression models (*n* = 736).

	Lead		Cadmium	
	Adjusted *β* (SE)	*p*	Adjusted *β* (SE)	*p*
**Lipid Profiles (mg/dL)**				
LDL-C	4.371 (0.654)	<0.001	5.748 (0.862)	<0.001
sdLDL-C	1.182 (0.228)	<0.001	1.225 (0.302)	<0.001
LDL-TG	1.811 (0.230)	<0.001	2.330 (0.559)	<0.001
HDL-C	−0.343 (0.202)	0.091	−0.395 (0.267)	0.140
Lipoprotein (a)	0.382 (0.231)	0.099	0.034 (0.306)	0.913
Apolipoprotein A1	0.350 (0.390)	0.370	0.438 (0.514)	0.394
Apolipoprotein B	0.338 (0.397)	0.395	−0.190 (0.523)	0.717
Triglyceride	1.123 (1.666)	0.501	−0.701 (2.197)	0.750
**CIMT (µm)**	14.939 (1.053)	<0.001	18.067 (1.418)	<0.001

Model: adjusted for age, sex, BMI z score, smoking status, drinking status, exercise, household income, dietary sweets, dietary fat, systolic blood pressure, HOMA-IR. Abbreviations: BMI: body mass index; HOMA-IR: homeostasis model assessment of insulin resistance; HDL-C: high-density lipoprotein cholesterol; LDL-C: low-density lipoprotein cholesterol; LDL-TG: low-density lipoprotein triglyceride; sdLDL-C: small dense low-density lipoprotein cholesterol.

**Table 3 nutrients-15-00571-t003:** Mean of CIMT, LDL-C, sdLDL-C, and LDL-TG in different lead and cadmium subgroups in the linear regression model.

		Pb ≤ 50th and Cd ≤ 50th *N* = 315	Pb > 50th and Cd ≤ 50th *N* = 55	Pb ≤ 50th and Cd > 50 *N* = 55	Pb > 50th and Cd > 50th *N* = 313
CIMT (μm)	Mean (S.E.)	435.08 (5.90)	438.74 (8.24)	441.36 (8.39)	477.33 (5.79)
	*p* value	Reference	0.601	0.372	<0.001
	*p* for trend	<0.001			
LDL-C (mg/dL)	Mean (S.E.)	76.66 (3.53)	73.31 (4.93)	79.69 (5.02)	89.79 (3.47)
	*p* value	Reference	0.425	0.470	<0.001
	*p* for trend	<0.001			
sdLDL-C (mg/dL)	Mean (S.E.)	14.27 (1.24)	13.60 (1.73)	14.70 (1.76)	16.85 (1.21)
	*p* value	Reference	0.644	0.772	0.002
	*p* for trend	0.008			
LDL-TG (mg/dL)	Mean (S.E.)	7.47 (2.29)	9.85 (3.20)	9.06 (3.26)	11.74 (2.25)
	*p* value	Reference	0.381	0.559	0.005
	*p* for trend	0.050			

Model: adjusted for age, sex, BMI z score, smoking status, drinking status, exercise, household income, dietary sweets, dietary fat, systolic blood pressure, and HOMA-IR. Abbreviations: BMI: body mass index; HOMA-IR: homeostasis model assessment of insulin resistance.

**Table 4 nutrients-15-00571-t004:** The relationship between Ln heavy metals (μg/g creatinine), lipoprotein profiles (mg/dL), and CIMT (μm) in the structural equation model.

Separate analysis	Lead→CIMT		Lead→lipoprotein profiles		Lipoprotein profiles→CIMT-	
	Adjusted β (SE)	*p* value	Adjusted β (SE)	*p* value	Adjusted β (SE)	*p* value
LDL-C	14.348 (1.012)	<0.001	4.370 (0.608)	<0.001	0.248 (0.059)	<0.001
sdLDL-C	14.607 (1.000)	<0.001	1.181 (0.213)	<0.001	0.657 (0.168)	<0.001
**Separate analysis**	**Cadmium→CIMT**		**Cadmium→lipoprotein profiles**		**Lipoprotein profiles→CIMT-**	
	**Adjusted β (SE)**	** *p* ** **value**	**Adjusted β (SE)**	** *p* ** **value**	**Adjusted β (SE)**	** *p* ** **value**
LDL-C	17.075 (1.372)	<0.001	5.719 (0.807)	<0.001	0.273 (0.061)	<0.001
sdLDL-C	17.597 (1.342)	<0.001	1.217 (0.284)	<0.001	0.806 (0.170)	<0.001
**Composite analysis**	**Lead→CIMT**		**Lead→lipoprotein profiles**		**Lipoprotein profiles→CIMT-**	
	**Adjusted β (SE)**	** *p* ** **value**	**Adjusted β (SE)**	** *p* ** **value**	**Adjusted β (SE)**	** *p* ** **value**
LDL-C	11.586 (0.987)	<0.001	2.467 (0.607)	<0.001	0.241 (0.059)	<0.001
sdLDL-C	11.223 (1.002)	<0.001	1.331 (0.213)	<0.001	0.666 (0.168)	<0.001
**Composite analysis**	**Cadmium→CIMT**		**Cadmium→lipoprotein profiles**		**Lipoprotein profiles→CIMT-**	
	**Adjusted β (SE)**	** *p* ** **value**	**Adjusted β (SE)**	** *p* ** **value**	**Adjusted β (SE)**	** *p* ** **value**
LDL-C	4.464 (1.306)	<0.001	3.022 (0.804)	<0.001	0.241 (0.059)	<0.001
sdLDL-C	5.386 (1.295)	<0.001	−0.238 (0.282)	0.399	0.666 (0.168)	<0.001

Model: adjusted for age, sex, BMI z score, smoking status, drinking status, exercise, household income, dietary sweets, dietary fat, systolic blood pressure, and HOMA-IR. Abbreviations: BMI: body mass index; HOMA-IR: homeostasis model assessment of insulin resistance; HDL-C: high-density lipoprotein cholesterol; LDL-C: low-density lipoprotein cholesterol; LDL-TG: low-density lipoprotein triglyceride; sdLDL-C: small dense low-density lipoprotein cholesterol.

**Table 5 nutrients-15-00571-t005:** The direct, indirect, and total effects (95% CI) of ln-heavy metals (μg/g creatinine) on CIMT (μm) when controlling the mediator of lipoprotein profiles (mg/dL) in the structural equation models.

	Lead→CIMT			Cadmium→CIMT		
	Total Effect	Direct Effect	Indirect Effect	Total Effect	Direct Effect	Indirect Effect
Separate analysis						
LDL-C	15.43 (13.62–18.28)	14.35 (12.24–16.78)	1.09 (0.41–1.78)	18.64 (15.70–21.97)	17.08 (14.12–20.13)	1.56 (0.69–2.57)
sdLDL-C	15.38 (13.50–18.22)	14.61 (12.45–16.81)	0.78 (0.03–1.71)	18.58 (15.60–21.88)	17.60 (14.44–20.81)	0.98 (0.13–2.39)
Composite analysis						
LDL-C	12.11 (7.97–15.47)	11.22 (7.08–14.94)	0.89 (0.13–2.44)	5.19 (0.52–10.28)	4.46 (0.17–9.65)	0.73 (0.05–1.83)
sdLDL-C	12.18 (7.95–15.40)	11.59 (7.32–14.91)	0.59 (0.13–1.49)	5.23 (0.68–10.31)	5.39 (0.99–10.37)	−0.16 (−0.22–0.56)

Model: adjusted for age, sex, BMI z score, smoking status, drinking status, exercise, household income, dietary sweets, dietary fat, systolic blood pressure, and HOMA-IR. Abbreviations: BMI: body mass index; HOMA-IR: homeostasis model assessment of insulin resistance; HDL-C: high-density lipoprotein cholesterol; LDL-C: low-density lipoprotein cholesterol; sdLDL-C: small dense low-density lipoprotein cholesterol.

## Data Availability

The datasets generated during and/or analyzed during the current study are available from the corresponding author on reasonable request.

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
