# Peer review of "Association of Urinary Lead and Cadmium Levels, and Serum Lipids with Subclinical Arteriosclerosis: Evidence from Taiwan"

_nutrients, 2023, doi:10.3390/nu15030571_

Round 1

Reviewer 1 Report

In this cross-sectional study, the authors aimed to explore the association between urine concentrations of lead and cadmium, serum lipoprotein markers and carotid intima media thickness (CIMT).

Although the manuscript addresses a relevant topic, there are major concerns to address.

Mayor Comments:

  1. Introduction section: An explanation or justification for the evaluation of these two heavy metals in particular (lead and cadmium) and not others (such as mercury, chromium or arsenic that are also associated with adverse health outcomes) is needed.
  2. Lines 53-54, Introduction section: Please change “endothelial-platelet microparticles, which are associated with endothelial damage” instead “endothelial-platelet microparticles, which represent vascular endothelial cell functions.
  3. Although the authors commented that the present study is in the framework of YOTA cohort, a brief description of the main characteristics of the participants is missing in the text.
  4. Line 40, Material and Methods: Please indicate which important variables were lacking in the patients.
  5. Please specify in the text whether the diabetes mellitus is type 1 or 2 or both.
  6. It is rather confusing to include heavy metals and CIMT in a section called "covariates" when both are the main targets of the article. On the other hand, it is not well explained in the text that the methodology of both (determination of lead and cadmium levels and CIMT) are detailed in the supplementary material.
  7. Lines 203-205: What do the authors mean by “Our study participants were based on those with abnormal urine tests in their childhood”? This fact is not previously explained in the text.
  8. Throughout the discussion, the authors make certain assumptions and hypotheses that are not consistent based on the different statistical analyses performed and that do not coincide, in part, with what has been previously described in the literature (page 5, lines 204-205; page 5, lines 240-243; page 6, lines 264-267).

MINOR COMMENTS:

  1. Page 2, Line 62: Please change ”treatment” by “exposure”.
  2. Page 3, Line 138: Please modify Table 3 by Table 2.
  3. Page 5, Line 255: Reference is not included in the correct format.

Reviewer 2 Report

It is undebatable that chronic metabolic diseases are induced by various factors. The manuscript by Lin et al. attempted to examine the association between urinary levels of potentially toxic trace metals (lead and cadmium), serum lipid profiles, and subclinical arteriosclerosis in a Taiwanese population. The report merit publication. I have the following minor comments on the draft.

1. Title

As the study recruited Taiwanese, with specific focus on two heavy metals (lead and cadmium), the title could preferably be revised to: Association of Urinary Lead and Cadmium Levels, and Serum Lipids with Subclinical Arteriosclerosis: Evidence from Taiwan.

2. Abstract

L28-29: It is important to indicate whether the association is significant or not.

L32: sdLDL-C refers to small dense LDL-C?

L34: may >> could

3. Keywords

Unusual abbreviations, and words already in the manuscript title are not good for listing as author-suggested indexing keywords. I suggest omitting abbreviations given, and also lipid profiles. Cardiovascular diseases, could be added as an author-suggested indexing keyword

4. Introduction

L40-45: These lines should be moved down, after cardiovascular diseases, which is the main issue addressed in this study.

5. Results

L430 (Table 1): I find it slightly inappropriate to indicate this as descriptive statistics. The data appears more of a demographic profile. Revise this to: Table 1. Demographic data and metal exposure variables in the studied Taiwanese population. In addition, there is repeated use of % symbol in this table. I suggest deleting all the others except the one in the row ''Numbers (%)''

6. Supplementary Materials

As a general guideline, the title page of the supplementary materials should be precisely the same as that of the main article. Similarly, referencing follows that of the main text. Please revise.

L48: Please delete ‘‘at’’.

The line numbers may be omitted in the final version to be published.

Round 2

Reviewer 1 Report

All comments have been properly addressed.

No more comments.

Author Response

Please see the attached file bellowed. Thanks!
